# Soybean Oil Epoxidation: Kinetics of the Epoxide Ring Opening Reactions

**Elio Santacesaria** [1],*[ID]**, Rosa Turco** [2][ID]**, Vincenzo Russo** [2][ID]**, Riccardo Tesser** [2] **and Martino Di Serio** [2]

1   CEO of Eurochem Engineering Ltd., 20139 Milano, Italy
2   NICL—Department of Chemical Science, University of Naples Federico II, 80126 Naples, Italy;
    rosa.turco@unina.it (R.T.); vincenzo.russo@unina.it (V.R.); riccardo.tesser@unina.it (R.T.);
    martino.diserio@unina.it (M.D.S.)
*   Correspondence: elio.santacesaria@eurochemengineering.com

**Abstract:** The epoxide ring opening reaction (ROR) can be considered as the most important side reaction occurring in the epoxidation of soybean oil reaction network. This reaction consistently reduces the selectivity to epoxidized soybean oil (ESBO). The reaction is also important for producing polyols and lubricants. In this work, the reaction was studied in different operative conditions to evaluate the effect on ROR rate respectively: (i) The Bronsted acidity of the mineral acid ($H_2SO_4$ or $H_3PO_4$), used as catalyst for promoting the oxidation with hydrogen peroxide of formic to performic acid, that is, the reactant in the epoxide formation; (ii) the concentration of the nucleophilic agents, normally present during the ESBO synthesis like HCOOH, HCOOOH, $H_2O$, $H_2O_2$; (iii) the stirring rate that changes the oil–water interface area and affects the mass transfer rate; (iv) the adopted temperature. Many different kinetic runs were made in different operative conditions, starting from an already epoxidized soybean oil. On the basis of these runs two different reaction mechanisms were hypothesized, one promoted by the Bronsted acidity mainly occurring at the oil–water interface and one promoted by the nucleophilic agents, in particular by formic acid. As it will be seen, the kinetic laws corresponding to the two mentioned mechanisms are quite different and this explain the divergent data reported in the literature on this subject. All the kinetic runs were correctly interpreted with a new developed biphasic kinetic model.

**Keywords:** epoxides; soybean oil; hydrogen peroxide; kinetics; ring opening reaction

## 1. Introduction

The epoxide ring opening reaction (ROR) of epoxidized vegetable oils has been intensively studied by many researchers interested in the production of polyols [1] or to the use as lubricants [2] or as intermediates for polyurethanes production [3]. In this case the scope is to find a good catalyst for promoting the reaction in a short time. On the contrary, in the epoxidation of vegetable oils (Prileschajew reaction [4]), ROR is an undesired side reaction lowering the yield in the production of epoxidized vegetable oils and the studies are, therefore, focused to hinder the reaction as much as possible. An example of epoxidation reaction network can be summarized as follows:

$$H_2O_2 \ + \ HCOOH \ \rightleftarrows \ HCOOOH \ + \ H_2O \tag{1}$$

$$>\!\!C\!=\!\!C\!<\ +\ \ HCOOOH\ \longrightarrow\ >\!\!C\!\!-\!\!C\!\!<\ +\ \ HCOOH \tag{2}$$

and according to our previous insights [5,6];

$$>\!\!C\!\!-\!\!C\!\!<\ +\ H^+\ \rightleftharpoons\ >\!\!C\!\!-\!\!C\!\!<\ \rightleftharpoons\ >\!\!C\!\!-\!\!C\!\!<\ \xrightarrow{+Nu}\ -\!\!C\!\!-\!\!C\!\!- \tag{3}$$

The sequence of reaction is the same if acetic acid is used instead of formic acid. As it can be seen, according to this mechanism, the oxirane cleavage step occurs in series with the epoxidation reaction and is promoted by a Bronsted acid environment. Considering that the reaction of formic to performic acid is catalyzed by mineral acids, such as sulfuric or phosphoric acid, the same catalyst seems to promote also the ring opening side reaction. It is important to point out that reaction (1) occurs in the aqueous phase, where the mineral acid is dissolved, while epoxide rings are dissolved in the oil phase. Therefore, it is reasonable to assume that reaction (3) occurs at the water/oil interface as suggested by us in two of our previous works [5,6] and also by other authors [7–11] Then, considering that reaction (3) is deleterious to obtain epoxidized products at high yield, a detailed study dealing with the kinetics of this reaction can be useful in order to find the best operative conditions and to minimize the negative influence of this reaction. Different studies have already been performed, on the subject, investigating the influence of several parameters, such as the pH, the temperature, and composition of the reaction mixture [7,12]. According to these works, at a given pH, both formic and performic acids (in alternative acetic and peracetic) hydrogen peroxide and water can influence the reaction rate. From the collected experimental data, different previously cited authors proposed a global third-order reaction rate expression (4) such as:

$$r\ =\ k\,c_{\text{Epox}}\,c_{\text{Nu}}^2 \tag{4}$$

At a first glance, it is hard to explain theoretically a second order for the nucleophilic reagents, for this reason a more reliable kinetic approach could be useful to better interpret the experimental data. Another important observation comes out from the literature analysis, that is, the pH of the reaction environment is a key feature in promoting the oxirane ring opening reaction. Moreover, according to the already mentioned authors [5–12] the ring opening reaction mainly occurs at the water–oil interface. Starting from these two observations, the reaction scheme (3) can be simplified by assuming the carbocation formation, following the protonic attack, as the rate determining step. On the basis of this conclusion, we can write:

$$>\!\!C\!\!-\!\!C\!\!<\ \rightleftharpoons\ >\!\!C\!\!-\!\!C\!\!< \tag{5}$$

Far from the equilibrium, the corresponding rate can be written as follows:

$$r\ =\ k\,c_{\text{Epox}^+} \tag{6}$$

But, the $c_{\text{Epox}^+}$ concentration, can be determined by considering the first reaction in (3) nearly at equilibrium. Therefore, we can write:

$$c_{\text{Epox}^+} = K_{\text{eq}}\, c_{\text{Epox}}\, c_{\text{H}^+} \tag{7}$$

Then, the resulting rate law becomes:

$$r = k\, K_{\text{eq}}\, c_{\text{Epox}}\, c_{\text{H}^+} = k_{\text{d}}\, c_{\text{Epox}}\, c_{\text{c}} \tag{8}$$

$k_{\text{d}}$, the overall pseudo-kinetic constant, contains also the reacting interfacial area changing according to the fluid dynamic conditions (for example the stirring speed). On the contrary, if the successive step is rate determining, that is, the nucleophilic attack to the protonated epoxide ring we can write:

$$r_{\text{Nu1}} = k_{\text{Nu1}}\, c_{\text{Epox}^+}\, c_{\text{Nu}} = k_{\text{Nu1}}\, K_{\text{eq}}\, c_{\text{Epox}}\, c_{\text{Nu}}\, c_{\text{H}^+} \tag{9}$$

Clearly, any nucleophilic component could have a different effect in the attack and therefore a different ring opening rate. In this case, the overall ring opening rate will result as the sum of different contributions to the reaction of respectively $H_2O$, HCOOH, HCOOOH, and $H_2O_2$.

When the aqueous solution, in contact with ESBO, contains a strong acid like $H_2SO_4$, the described mechanism and kinetics seems the most reliable, but in less acid environment also another reaction mechanism could become competitive, that is, a direct nucleophilic attack to one of the two carbon atoms of the oxirane ring, the formation of an intermediate with a negative charge and the abstraction of a proton from another nucleophilic molecule, such as:

$$\tag{10}$$

In this case, the reaction rate becomes independent of the protonic concentration but has a dependence on the square concentration of the nucleophilic molecules, that is:

$$r_{\text{Nu2}} = k_{\text{Nu2}}\, c_{\text{Epox}}\, c_{\text{Nu}}^2 \tag{11}$$

Also in this case the nucleophilic power of the involved molecules (reactants and products) can be different and the overall ROR rate, occurring with this mechanism, is again the sum of the contributions of the different involved components ($H_2O$, HCOOH, HCOOOH, and $H_2O_2$). In conclusion, in the most general cases, the overall ring opening rate will be the sum of the two contributions $r_{\text{Nu1}} + r_{\text{Nu2}}$ related to the occurrence of respectively the two mentioned different mechanisms.

In this work, starting from an already epoxidized soybean oil, many different kinetic runs have been performed in the presence of different concentrations of mineral acids such as $H_2SO_4$ and $H_3PO_4$, in the absence of mineral acids (auto-catalysis), in the presence of different concentrations of all the possible nucleophilic agents, in particular: HCOOH, HCOOOH, $H_2O_2$, and $H_2O$. Moreover, also the effect of both the stirring rate (interface area) and the temperature on the reaction rate has been verified. Almost all the runs have successfully been simulated with the previously described kinetic laws and the related parameters have been determined by mathematical regression analysis. A kinetic model has been developed taking into account, when necessary, also the reactions of performic acid formation and decomposition, occurring according to the following two reactions [13]:

$$H_2O_2 \ + \ HCOOH \ \rightleftarrows \ HCOOOH \ + \ H_2O \tag{12}$$

$$HCOOOH \ \rightarrow \ CO_2 \ + \ H_2O \tag{13}$$

This last reaction consumes directly the reactant HCOOH and indirectly $H_2O_2$, therefore, this must be considered when ROR is performed in the presence of both formic acid and hydrogen peroxide, because, the composition of the aqueous solution changes along the time. The kinetics of these reactions have been studied in detail in a previously published work [13] and the kinetic laws and related parameters reported there have been employed in the present ROR model.

At last, another aspect arising in the ROR kinetic interpretation is the role of mass transfer, considering that epoxidized soybean oil is a viscous liquid and the epoxidized molecules are bulky molecules that diffuse slowly from the oil bulk to the interface. As it will be seen, also this aspect has been considered in the developed kinetic model.

## 2. Experimental Section

*Materials*

Epoxidized soybean oil (ESBO), with an oxirane number of 6.58 (grams of oxygen for 100 g of sample) and an iodine value of 2.0 (grams of reacted iodine per 100 g of sample) was purchased by KCHIMICA S.r.L. Hydrogen peroxide (60%) was supplied by Solvay Italia S.p.A. Formic acid (96%), sulfuric acid (97%), phosphoric acid (85%), all numbers are given as mass fraction %. All other employed reagents were provided by Merck at the highest level of purity available (>99.9%) and were used as received without further purification.

## 3. Apparatus

The kinetic runs have been carried out in a well stirred cylindrical jacketed glass reactors (purchased by Vetrochimica Srl, Casandrino (Naples), Italy) of 500 cm$^3$ of volume and equipped with both a thermocouple for the reaction temperature control and a magnetically driven stirrer. The temperature control in the reactor was made by using recirculating thermostatted water.

## 4. Methods

The ring opening runs were carried out by working with aqueous solution of $H_2O_2$, water, and formic acid in the presence of sulfuric acid, phosphoric acid or in the absence of mineral acids. Degradation tests were carried out in the experimental conditions normally reached in the final part (digestion phase) of the soybean oil epoxidation process, where hydrogen peroxide and sulfuric or phosphoric acids are more diluted. In particular, a classical experimental procedure is described as it follows: a mixture containing 36.7 g of hydrogen peroxide (with a mass fraction of 20%), 5.4 g of formic acid (mass fraction 95%), and 0.64 g of sulfuric acid (mass fraction 98%) or in alternative phosphoric acid (mass fraction 85%), was added to 100 g of well mixed epoxidized soybean oil, heated at a desired temperature (normally 70 °C). As soon as the substrate reached the established temperature, the aqueous mixture was fed to the reactor (time = 0) and samples were periodically withdrawn during the 4–5 h of reaction. Then, the withdrawn samples have been cooled, a sodium bicarbonate solution was added to neutralize the residual acidity and finally anhydrous magnesium sulfate was added to remove the water. The oil of the withdrawn samples was analyzed for determining the oxirane number (ON), according to an analytical method reported by the literature [14–16]. The average titration error is inferior to 0.6%. A list of the experimental conditions for all the performed degradation runs are detailed in Table 1.

**Table 1.** List of ring opening experimental runs performed. 100 g of ESBO was used for all the runs. rpm = number of stirrer revolutions per minute. w = mass fraction %.

| Run | Type Catalyst | Catalyst (g) | Temperature (°C) | Stirring Rate (rpm) | HCOOH (w = 96%) (g) | $H_2O_2$ (w = 20%) (g) |
|---|---|---|---|---|---|---|
| 1 | | 0.65 | 70 | 750 | 5.38 | 36.7 |
| 2 | | 0.65 | 70 | 1 000 | 5.38 | 36.7 |
| 3 | | 0.65 | 70 | 500 | 5.38 | 36.7 |
| 4 | | 0.65 | 70 | 250 | 5.38 | 36.7 |
| 5 | | 0.65 | 80 | 750 | 5.38 | 36.7 |
| 6 | $H_2SO_4$ | 0.65 | 90 | 750 | 5.38 | 36.7 |
| 7 | (w = 97%) | 1.29 | 70 | 750 | 5.38 | 36.7 |
| 8 | | 0.32 | 70 | 750 | 5.38 | 36.7 |
| 9 | | 0.65 | 70 | 750 | 5.38 | 36.7 |
| 10 | | 0.65 | 70 | 750 | 5.38 | 36.7 |
| 11 | | 0.65 | 70 | 750 | 10.76 | 36.7 |
| 12 | | 0.65 | 70 | 750 | 2.69 | 36.7 |
| 13 | | 0.65 | 70 | 250 | 5.16 | 36.7 |
| 14 | | 0.65 | 70 | 500 | 5.16 | 36.7 |
| 15 | | 0.65 | 70 | 750 | 5.16 | 36.7 |
| 16 | $H_3PO_4$ | 1.30 | 70 | 500 | 5.16 | 36.7 |
| 17 | (w = 85%) | 0.79 | 70 | 500 | 5.16 | 36.7 |
| 18 | | 0.65 | 70 | 500 | 10.33 | 36.7 |
| 19 | | 0.65 | 60 | 500 | 5.16 | 36.7 |
| 20 | | 0.65 | 80 | 500 | 5.16 | 36.7 |
| 21 | | 0.65 | 70 | 500 | 5.16 | 36.7 [a] |
| 22 | | - | 70 | 750 | 4.25 | $H_2O$ 1.13 g [b] |
| 23 | None | - | 70 | 750 | 4.25 | 7.34 |
| 24 | | - | 70 | 750 | 4.25 | $H_2O$ 38.47 [b] |

[a]: Hydrogen peroxide 30 wt.% was used for this run. [b]: No $H_2O_2$ added but different amounts of water.

## 5. Results and Discussion

### 5.1. Runs Performed in the Presence of $H_2SO_4$

Twelve kinetic runs were performed in the presence of $H_2SO_4$ to evaluate the effect on the epoxide ROR rate of the following factors: (1) stirring rate (Interfacial area); (2) content of sulfuric acid; (3) temperature; (4) content of formic acid.

The list of the experimental runs performed in the presence of sulfuric acid as catalyst is reported on Table 1.

### 5.2. Stirring Rate Effect

In order to investigate the effect of the interfacial area, some runs, at different stirring rates, were performed, as it can be seen in Table 1. In Figure 1 the profiles along the time of the oxirane number, obtained at different stirring rates, are reported. From this figure it is evident that, in the presence of sulfuric acid (a strong Bronsted acid), by increasing the stirring rate, that is, by increasing the liquid–liquid interfacial area, the ring-opening reaction rate increases, too. This behavior means that reaction rates, in this case, are more or less affected by the mass transfer, the migration of the epoxide molecules from the bulk of the oil to the oil-water reacting interface being relatively slow. This aspect is quantitatively interpreted by the developed kinetic model that is described later.

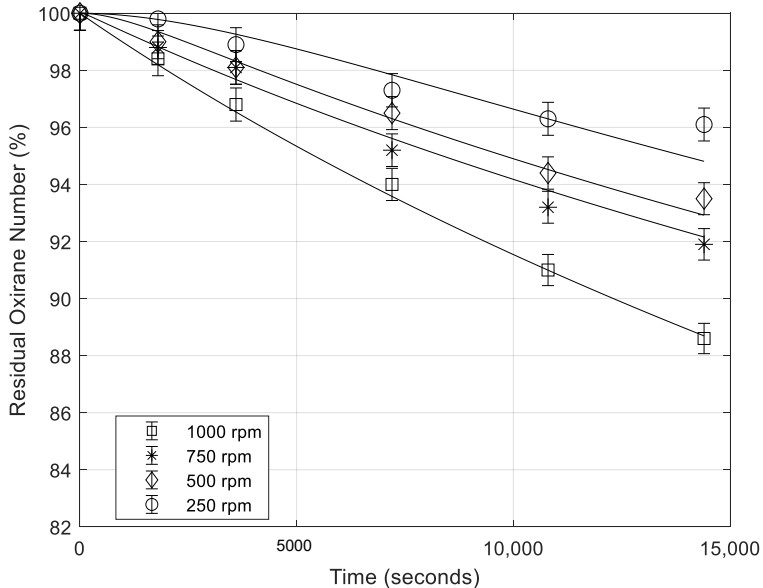

**Figure 1.** Residual oxirane number profiles vs. time for different stirring rate (rpm = rotations per minute). Points are experimental data lines that are calculated with the developed model (mass of catalyst 0.65 g; temperature 70 °C; mass of formic acid 5.38 g).

### 5.3. Effect of $H_2SO_4$ Concentration

In order to investigate the effect of the catalyst content, runs at different $H_2SO_4$ concentrations were performed, as it can be seen in Table 1. In Figure 2, the oxirane number profiles decreasing for the degradation reaction as a consequence of different catalyst concentrations are reported. As it can be seen, $H_2SO_4$ concentration has a great influence on the degradation rate, but the behavior is not linear.

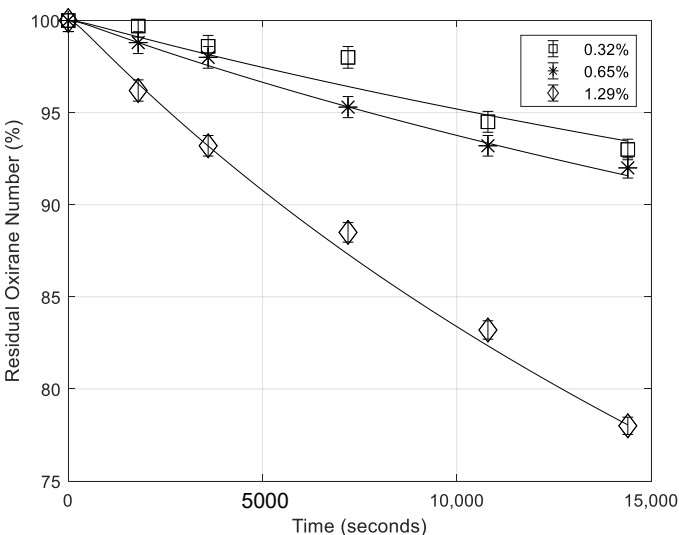

**Figure 2.** Residual oxirane number profiles vs. time for different sulfuric acid concentrations. Points are experimental data lines are calculated with the developed model. (temperature 70 °C; rate of agitation 750 rpm (rpm = revolutions per minute); mass of formic acid 5.38 g).

### 5.4. Effect of the Temperature

Runs at different temperatures were performed, taking the other conditions the same as before. In Figure 3, the obtained results are reported as profiles of the oxirane numbers, decreasing for the degradation reaction, at different temperatures.

As it is can be seen, temperature has a great influence on the rate of the epoxide ring opening, in particular on the initial reaction rate.

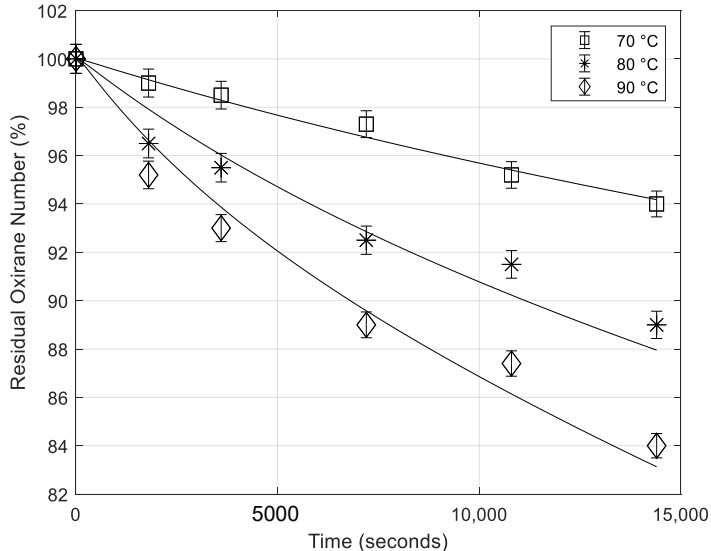

**Figure 3.** Residual oxirane number profiles obtained at different temperatures. Points are experimental data lines that are calculated with the developed model (mass of catalyst 0.65 g; rate of agitation 750 rpm; mass of formic acid 5.38 g).

## 5.5. Effect of Formic Acid Concentration

The profiles of oxirane numbers obtained for different initial formic acid concentrations, keeping constant the other conditions, are reported in Figure 4. As it can be observed also formic concentration has a relevant effect on the degradation rate of ESBO.

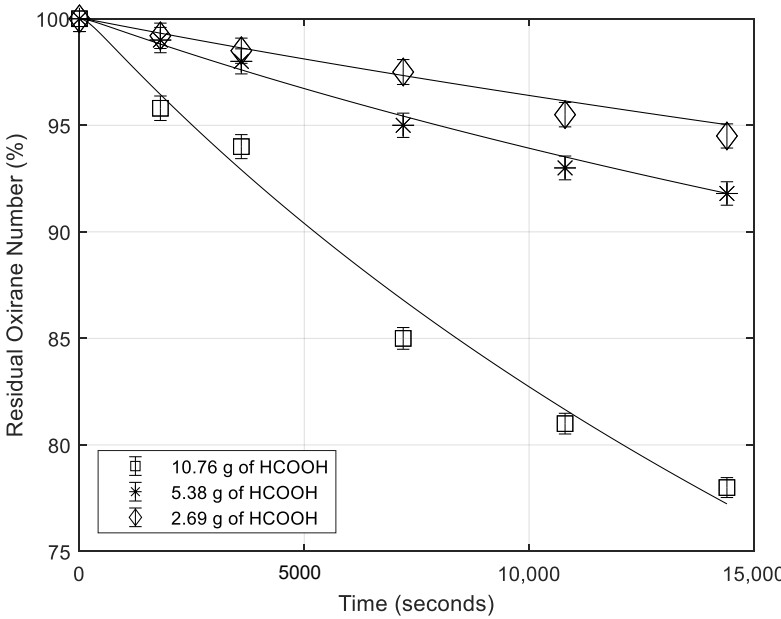

**Figure 4.** Residual oxirane number profiles obtained for different initial concentrations of formic acid. Points are experimental data lines that are calculated with the developed model (mass of catalyst 0.65 g; temperature 70 °C; rate of agitation 750 rpm).

*5.6. Remarks on the Kinetic Results Obtained in the Presence of Sulfuric Acid*

Figures 1 and 2 confirm that in the presence of a strong Bronsted acid, like sulfuric acid, the ROR occurs mainly at the liquid–liquid interface, because, the reaction rate is very sensible to the stirring rate. The effect of $H_2SO_4$ concentration seems not linear, because, a small difference has been observed by doubling the concentration at the lower levels and a threshold limit seems to occur at higher concentration. These findings are in agreement with Cai et al. [17], although their investigation was related to a different oil (cottonseed) and the use of acetic instead of formic acid.

As expected, the effect of the temperature is significant, while, it was not expected the remarkable influence of formic acid probably acting as nucleophilic agent. This means that our previously suggested mechanism (see relation 5) is not respected and the rate determining step seems to be the nucleophilic attack to the protonated oxirane ring. This would correspond to a kinetic law of the type:

$$r = k_{Nu} c_{Epox^+} c_{Nu} \tag{14}$$

$$\text{but } c_{Epox^+} = K_{eq} c_{Epox} c_{H^+} \tag{15}$$

$$\text{therefore, } r = k_{Nu} K_{eq} c_{Epox} c_{H^+} c_{Nu} \tag{16}$$

where $c_{Nu}$ corresponds to the cumulative concentration of all the nucleophilic agents present in the reacting mixture. Therefore, considering that each nucleophilic agent acts with a different power, we can differentiate their contribution by writing:

$$r_{10} = k_{10} c_{Epox} c_{H^+} \left( \alpha_1 c_{FA} + \alpha_2 c_{H_2O_2} + \alpha_3 c_{PFA} + \alpha_4 c_{H_2O} \right) \left( mol/cm^3 \, s \right) \tag{17}$$

The reaction rate is referred to the oil volume unit expressed in $cm^3$.

Clearly, all these observations must be demonstrated through an opportune kinetic model able to simulate all the performed runs.

## 6. Runs Performed in the Presence of $H_3PO_4$

The kinetic runs, performed in the presence of $H_3PO_4$, have been made according to the experimental conditions as reported in Table 1. The following variables have been considered, in this case: (1) the stirring rate; (2) the content of phosphoric acid; (3) the temperature; (4) the content of formic acid; (5) the content of hydrogen peroxide.

*6.1. Stirring Rate Effect*

In order to investigate the effect of the interfacial area, runs at different stirring rates were performed (see in Table 1 the adopted experimental conditions). In Figure 5 the oxirane number profiles, obtained for different stirring rates are reported. As it can be seen, in the presence of $H_3PO_4$, this factor is poorly influent. The simulation of the different runs reported in the figure has been obtained with just a small change of the mass transfer coefficient $\beta$.

*6.2. Effect of $H_3PO_4$ Concentration*

Some runs have been performed in the presence of different $H_3PO_4$ concentrations (see Table 1). The obtained results are reported in the usual way in Figure 6. As it can be seen, also the phosphoric acid concentration poorly affects the degradation rate, probably, because the pH changes little by changing phosphoric acid concentration.

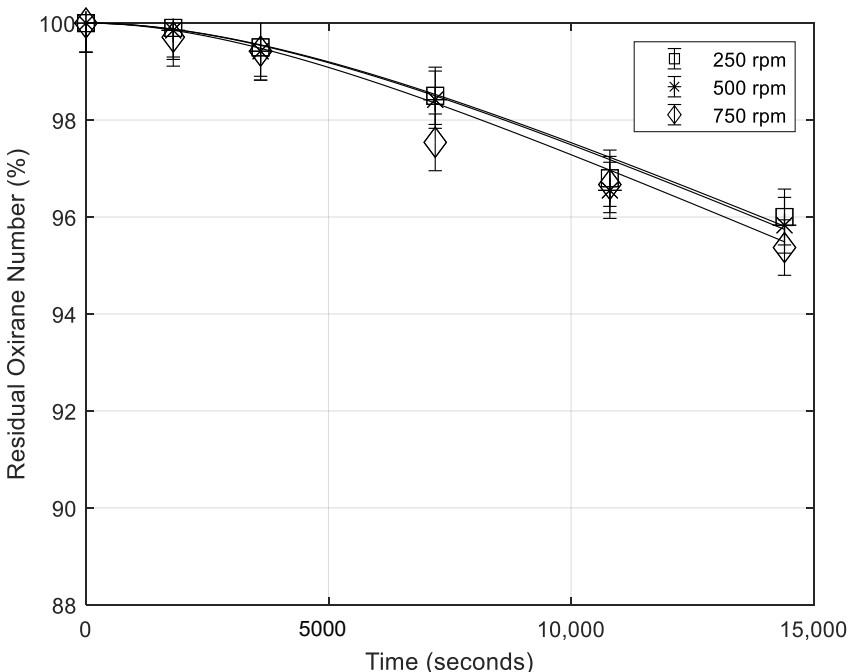

**Figure 5.** Residual oxirane number profiles in the presence of the same amount of $H_3PO_4$ for different stirring rates (rpm = number of stirrer revolution per minute). Points are experimental data lines that are calculated with the developed model (mass of catalyst 0.65 g; temperature 70 °C; mass of formic acid 5.16 g).

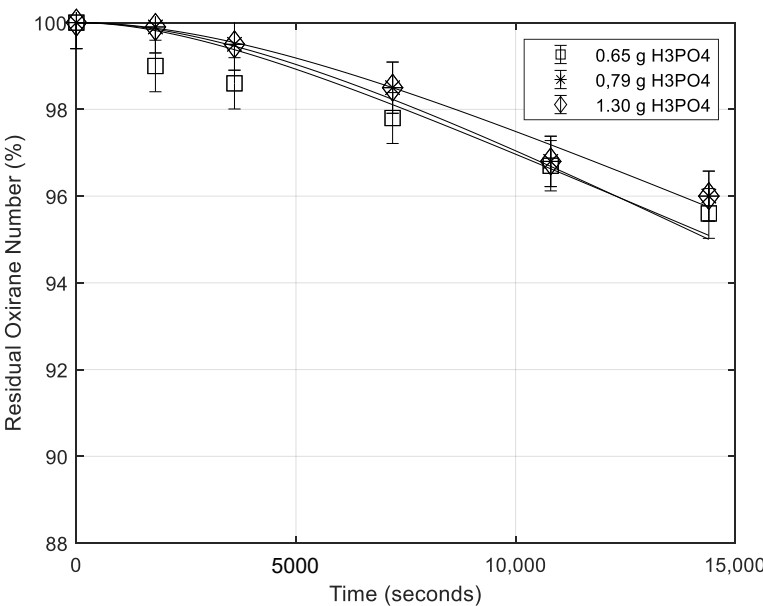

**Figure 6.** Residual oxirane number profiles for different contents of $H_3PO_4$. All the other conditions are the same (temperature 70 °C; rate of agitation 500 rpm; mass of formic acid 5.16 g). Points are experimental data lines that are calculated with the developed model.

## 6.3. Effect of the Temperature

The effect of the temperature in the presence of $H_3PO_4$ catalyst is similar to the one observed for $H_2SO_4$ as it can be appreciated in Figure 7.

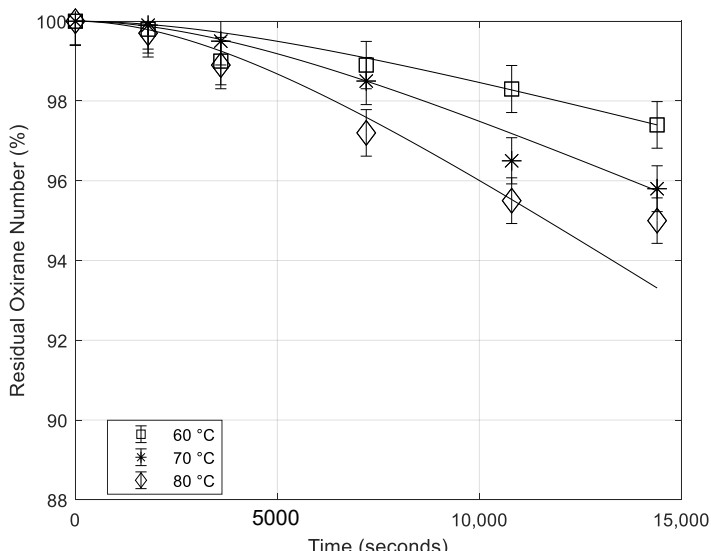

**Figure 7.** Residual oxirane number profiles for different temperatures. All the other conditions are the same (mass of catalyst 0.65 g; rate of agitation 500 rpm; mass of formic acid 5.16 g). Points are experimental data lines that are calculated with the developed model.

*6.4. Effect of Formic Acid Content*

The oxirane number profiles for different formic acid concentration are reported in Figure 8. Again, a significant effect of the HCCOH concentration on the ESBO degradation rate has been observed.

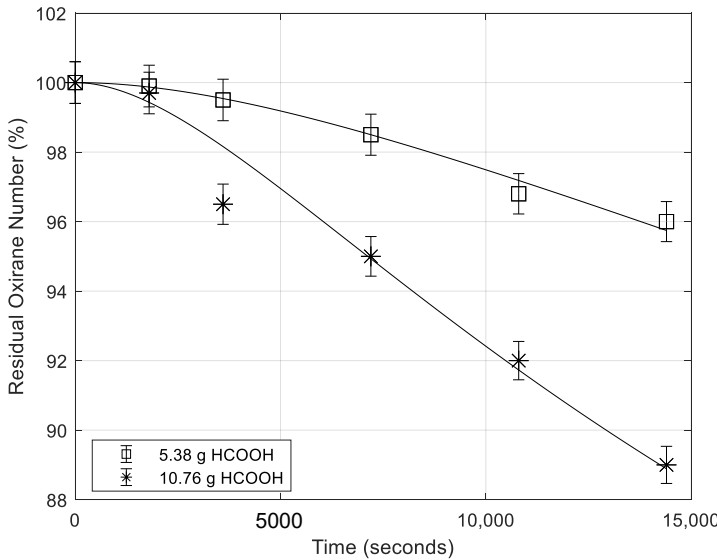

**Figure 8.** Residual oxirane profiles for different contents of formic acid being the same for all the other conditions (mass of catalyst 0.65 g; temperature 70 °C; rate of agitation 500 rpm). Points are experimental data lines that are calculated with the developed model.

*6.5. Effect of Hydrogen Peroxide Concentration*

The effect of hydrogen peroxide concentration on the ROR rate is moderate (see Figure 9), probably affecting the rate by oxidizing formic to performic acid being this last less active as nucleophilic agent. Therefore, the amount of formic acid in the reaction mixture is reduced.

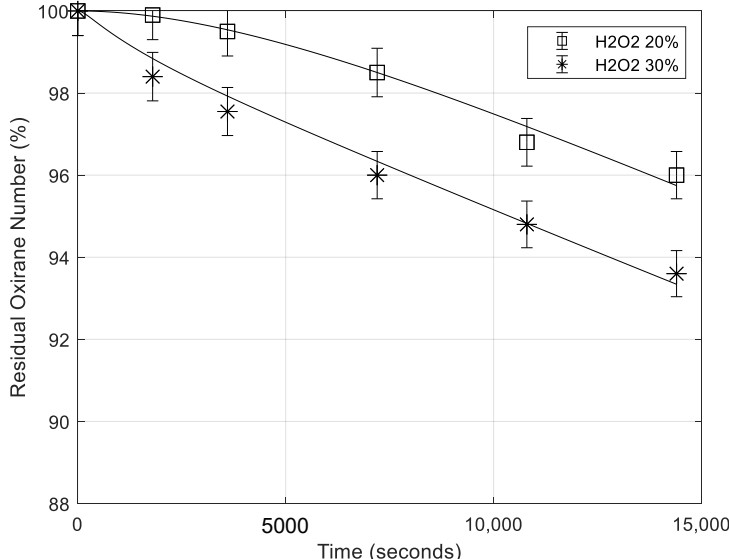

**Figure 9.** Residual oxirane profiles for different concentrations of hydrogen peroxide, being the same all the other conditions (mass of catalyst 0.65 g; rate of agitation 500 rpm; temperature 70 °C, mass of formic acid 5.16 g). Points are experimental data lines that are calculated with the developed model.

*6.6. Remarks on the Kinetic Results Obtained in the Presence of $H_3PO_4$ Catalyst*

We observed a great difference in the runs performed in the presence of $H_3PO_4$ with respect to the ones made in the presence of $H_2SO_4$, in particular for what concerns the effect of stirring rate. While, in the presence of sulfuric acid the effect of the interface area was significant, and in the case of phosphoric acid a very small influence was observed, this behavior can be explained by assuming that phosphoric acid strongly interacts with the oxygen of the epoxide ring stabilizing it and giving place to a complex having good surfactant properties. In this case, a high interface area is formed also at low stirring rates but this is not followed by an increase in the degradation rate.

Another difference observed is the very low influence of the phosphoric acid concentration on the epoxide ring opening rate, although this behavior could be in agreement with the small influence of sulfuric acid concentration that has been observed at low concentration levels.

However, the kinetic behavior of ROR in the presence of $H_3PO_4$ suggests the prevalence of the already suggested mechanism (10), characterized by the attack of a nucleophilic anion to one of the two carbon atoms of the epoxide ring followed by a proton abstraction by the negatively charged oxygen. A mechanism of this type is in agreement with the kinetic law previously reported in relation (11) and with the observation made by different authors [7,8,11–13]. We can write now relation (11) in a more detailed way as:

$$r_{11} = k_{11} c_{\text{Epox}} \left( \sum_i \gamma_i c_{\text{Nu}_i}^2 \right) (\text{mol}/(\text{cm}^3 \text{ s})) \tag{18}$$

As for $r_{10}$ the reaction rate is referred to the oil volume unit expressed in cm$^3$.

Clearly, it cannot be excluded, in certain operative conditions, the intervention of both the described mechanisms, in this case the ROR reaction rate becomes the sum of the two contributions $r_{10}$ and $r_{11}$.

Then, for what concerns the influence of formic acid, one of the four nucleophilic components, the behavior can be considered similar to the one observed for sulfuric acid but with a somewhat lower activity, confirming the protective effect of phosphoric acid molecule on the stability of the epoxide rings. Hydrogen peroxide seems to have an indirect effect on ROR rate by converting formic acid into performic acid, hence reducing its concentration. Finally, a relevant effect of the temperature has been observed.

## 7. Runs Performed in the Absence of Mineral Acids

Some runs have been performed in the absence of mineral acid with the scope to evaluate separately the contribution of formic acid or of the other nucleophilic components to the ring opening reaction. In Table 1, the runs performed, at this purpose, are summarized. In Run 22 of Table 1 only formic acid has been put in contact with ESBO. The obtained ESBO degradation for this run is shown in Figure 10. As it can be seen, the effect of concentrated formic acid is considerable in promoting the oxirane ring degradation rate, in agreement with the large effect previously observed by changing the HCOOH concentration in the presence of mineral acids. Run 23 was performed in absence of mineral acids but in the presence of both formic acid and hydrogen peroxide, which means to operate in the presence of three components and the third one being performic acid formed in situ. The obtained results are shown in Figure 10 for a useful comparison. By comparing these two runs it can be observed that in the last case the rate of oxirane ring opening is much slower probably because HCOOH is more diluted and consistently substituted by the performic acid which is much less acidic and less active as nucleophile. On the other hand, the effect of HCOOH dilution is again confirmed with run 24 always reported in Figure 10, occurring in the presence of a consistent amount of water.

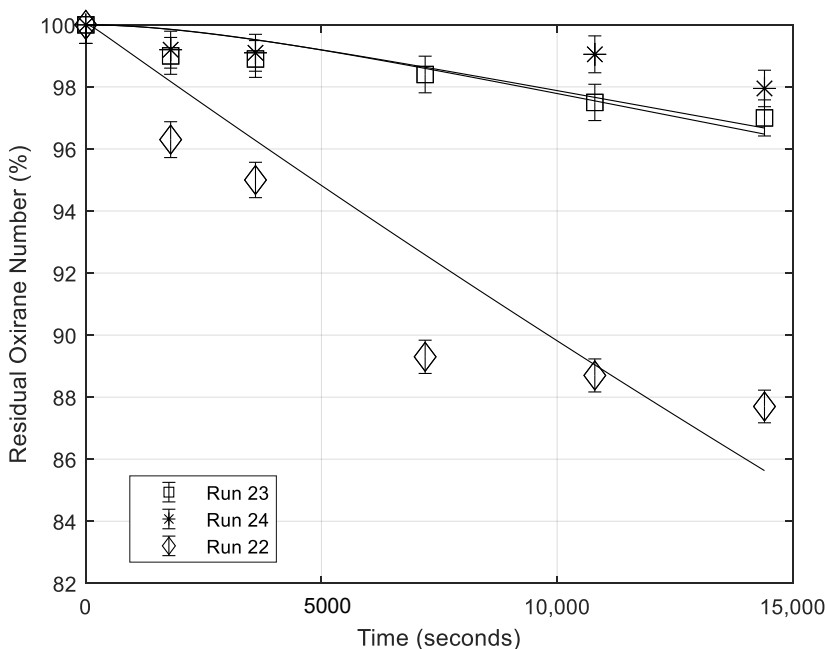

**Figure 10.** Evolution with time of the residual oxirane number in the presence of HCOOH alone (run 22), at much lower concentration because diluted with water (run 24), in the presence of $H_2O_2$, and performic acid (run 23).

## 8. Elaboration of the Experimental Kinetic Data of Ring Opening Reactions (ROR)

*Description of the Adopted Kinetic Model*

The selectivity of the soybean oil epoxidation is significantly lowered by the occurrence of some undesired side reactions characterized by the oxirane ring opening. These reactions become important in particular during the maturation step of the epoxidation process that has the scope of reaching the target values of both the oxirane number (greater than 6.5) and the iodine number (lower than 1–1.5). This phase of the ESBO production requires a long reaction time, at relatively high temperature (70–75 °C). Moreover, consider also that the epoxidation rate decreases with the double bonds disappearance, while, the ring opening reaction rate increases for the increase of the oxirane ring concentration. During this phase of the industrial process, an aqueous solution containing: HCOOH,

HCOOOH, $H_2O_2$ and $H_2O$, and $H_2SO_4$ or $H_3PO_4$ remains in contact with the epoxidized oil for at least 2–3 h. During this time epoxidation reaction continues consuming HCOOOH and indirectly $H_2O_2$ but, as we have previously seen, HCOOOH also decomposes to $CO_2$ and $H_2O$, further decreasing the $H_2O_2$ concentration and decreasing the HCOOH concentration, too. Therefore, the kinetic runs of the ring opening reaction (ROR) have been made by putting in contact an already epoxidized soybean oil with an aqueous solution of a composition similar to the one that we can found during the maturation phase of the process. As seen, runs have been made in the presence of sulfuric acid, phosphoric acid and in the absence of mineral acids to evaluate the role of both the solution acidity and the nucleophilic components concentrations. All the runs made have been simulated with a unique kinetic model and the parameters giving the best agreement with the experimental data have been determined. The model is based on the solution of the following differential equations system:

$$\frac{dc_{FA}}{dt} = -r_1 \text{ Rate of formic acid transforming in HCOOOH (mol/(L s))} \tag{19}$$

$$\frac{dc_{H_2O_2}}{dt} = -r_1 \text{ Rate of } H_2O_2 \text{ consumption to give HCOOOH (mol/(L s))} \tag{20}$$

$$\frac{dc_{PFA}}{dt} = r_1 - r_2 \; (r_1 = \text{PFA formation } r_2 = \text{PFA decomposition}) \text{ (mol/(L s))} \tag{21}$$

$$\frac{dc_{H_2O}}{dt} = r_1 + r_2 \text{ Rate of water formation (mol/L s)} \tag{22}$$

$$\frac{dc_{ESBO-bulk}}{dt} = -r_t \text{ Rate of ESBO molecules mass transfer} \tag{23}$$

$$\frac{dc_{ESBO-interface}}{dt} = r_t - r_g \text{ ESBO accumulation at the interface} \tag{24}$$

where, we have:

$$r_1 = k_1 \, c_{H^+} \, c_{FA} \, c_{H_2O_2} \left( 1 - \frac{1}{K_{eq}} \frac{c_{PFA} \, c_{H_2O}}{c_{FA} \, c_{CO_2}} \right) \tag{25}$$

$$r_2 = k_2 \, c_{PFA} \tag{26}$$

The rates of these reactions occurring in the aqueous phase are referred to as the aqueous volume unit expressed in liter (L).

$$K_{eq} = K_{eq,ref} \, \exp\left[ \left( \frac{-\Delta H}{R} \right) \cdot \left( \frac{1}{298} - \frac{1}{T} \right) \right]$$
$$= (1.6 \pm 0.1) \exp\left[ \left( \frac{10{,}000 \pm 900}{8.314} \right) \cdot \left( \frac{1}{298} - \frac{1}{T} \right) \right] \tag{27}$$

The dimensions are $k_1$ ($L^2/(mol^2 \, s)$) and $k_2$ (L/s). Equations (25)–(27) are relationships determined in a previous work published by the same authors[6]. $r_1$ being the rate of performic acid formation and $r_2$ the decomposition rate, the volume is referred to as the aqueous phase.

The kinetic parameters reported in the already mentioned work[6] are somewhat different in the presence of sulfuric acid, phosphoric acid or in the absence of mineral acids as it can be seen in Table 2. On the contrary, the equilibrium constant is obviously independent of the employed catalyst and are reported in relation (27).

**Table 2.** Kinetic parameters for the formation and decomposition of performic acid in different environment taken from [13]. Here, 323 K has been taken as the reference temperature.

| Catalyst | $k_{1\text{-}323}$ ($L^2/mol^2$ s) | $k_{2\text{-}323}$ (L/s) | $\Delta E_1$ (J/mol) | $\Delta E_2$ (J/mol) |
|---|---|---|---|---|
| $H_2SO_4$ | $(1.03 \pm 0.05) \times 10^{-3}$ | $(9.61 \pm 1.56) \times 10^{-5}$ | $47{,}518 \pm 3061$ | $87{,}497 \pm 2374$ |
| $H_3PO_4$ | $(1.16 \pm 0.13) \times 10^{-3}$ | $(1.36 \pm 0.32) \times 10^{-4}$ | $83{,}816 \pm 2905$ | $98{,}989 \pm 3014$ |
| NONE | $(1.20 \pm 0.16) \times 10^{-3}$ | $(1.60 \pm 0.32) \times 10^{-4}$ | $55{,}304 \pm 1288$ | $105{,}073 \pm 8419$ |

The term $r_g$ is the overall ring opening reaction rate considering as the sum of the contribution of the two described mechanisms, that is:

$$r_g = (r_{10} + r_{11}) \left(\text{mol/cm}^3 \text{ s}\right) \tag{28}$$

$$r_{10} = k_{10} \, c_{\text{Epox}} \, c_{\text{H}^+} \left(\alpha_1 \, c_{\text{FA}} + \alpha_2 \, c_{\text{H}_2\text{O}_2} + \alpha_3 c_{\text{PFA}} + \alpha_4 c_{\text{H}_2\text{O}}\right) \left(\text{mol}/\left(\text{cm}^3 \text{ s}\right)\right) \tag{29}$$

$r_{10}$ is the ring opening rate promoted by the acid environment in which the rate-determining step is the reaction of the protonated ring reacting with the nucleophilic agent. The parameters from $\alpha_1$ to $\alpha_4$ differentiate the contribution given by each nucleophilic molecule.

$$r_{11} = k_{11} \, c_{\text{Epox}} \left(\gamma_1 c_{\text{FA}}^2 + \gamma_2 \, c_{\text{H}_2\text{O}_2}^2 + \gamma_3 c_{\text{PFA}}^2 + \gamma_4 \, c_{\text{H}_2\text{O}}^2\right) \left(\text{mol/cm}^3 \text{ s}\right) \tag{30}$$

$r_{11}$ is the ring opening rate directly promoted by the nucleophilic agents. The parameters from $\gamma_1$ to $\gamma_4$ differentiate the contribution of each nucleophilic molecule.

Both $r_{10}$ and $r_{11}$ are referred to the oil volume unit expressed in cm$^3$. $k_{10}$ and $k_{11}$ dimensions are both (l$^2$/(mol$^2$ s)). Finally, $r_t$ is the mass transfer rate of the ESBO molecules from the liquid bulk to the interface where the ring opening reactions occur.

$$r_t = \beta \, (c_e - c_f) \left(\text{mol}/\left(\text{cm}^3 \text{ s}\right)\right) \tag{31}$$

The H$^+$ concentration changes more or less according to the acid used as catalyst and the temperature and for a correct simulation we have to evaluate such concentration.

## 9. Determination of the Hydrogen Ions Concentration at Different Temperature

As we have seen, $r_1$ and $r_{10}$ are both promoted by the acid environment. A rigorous approach would be to calculate what is the H$^+$ concentration at the reaction temperature and how the H$^+$ concentration changes along the time in the aqueous reactants by adding a solution of hydrogen peroxide and formic acid to the acid catalyst. At this purpose, we have developed three different program codes for solving the set of algebraic equations (multiple protonation equilibria) for determining respectively the H$^+$ concentration in a solution of formic acid (the solution to be added), in a solution of concentrated sulfuric acid or phosphoric acid (solutions in contact with the oil), and in a solution containing both formic and sulfuric or phosphoric acids (solution formed by adding the oxidant mixture to oil and catalyst). Before considering the three mentioned cases, in Table 3, the dissociation constants are reported at 25 °C for all the acids involved in the reaction.

**Table 3.** Ionic dissociation constants at 25 °C.

| Type of Dissociation | Dissociation Constant at 25 °C and Ionic Product |
|---|---|
| $\text{HCOOH} \rightarrow \text{H}^+ + \text{HCOO}^-$ | $1.77 \times 10^{-4}$ |
| $\text{HCOOOH} \rightleftarrows \text{H}^+ + \text{HCOOO}^-$ | $7.90 \times 10^{-8}$ |
| $\text{H}_3\text{PO}_4 \rightleftarrows \text{H}^+ + \text{H}_2\text{PO}_4{}^-$ | $7.11 \times 10^{-3}$ |
| $\text{H}_2\text{PO}_4{}^- \rightleftarrows \text{H}^+ + \text{HPO}_4{}^=$ | $6.31 \times 10^{-8}$ |
| $\text{HPO}_4{}^= \rightleftarrows \text{H}^+ + \text{PO}_4{}^{3-}$ | $4.80 \times 10^{-13}$ |
| $\text{H}_2\text{SO}_4 \rightleftarrows \text{H}^+ + \text{HSO}_4{}^-$ | $2.40 \times 10^6$ |
| $\text{HSO}_4{}^- \rightleftarrows \text{H}^+ + \text{SO}_4{}^=$ | $1.20 \times 10^{-2}$ |
| $\text{H}_2\text{O} \rightleftarrows \text{H}^+ + \text{OH}^-$ | $K_w = c_{\text{H}^+} \, c_{\text{OH}^-} = 10^{-14}$ |

## 10. Hydrogen Ion Concentration in the System Formic–Performic Acid

In the absence of mineral acids the acidity is only due to the dissociation of formic acid:

$$\text{HCOOH} \rightleftarrows \text{H}^+ + \text{HCOO}^- \tag{32}$$

The dissociation constant of performic acid is too low and therefore can be neglected.

Hence, we can write:

$$K_{aFA} = \frac{c_{H^+}\, c_{HCOO^-}}{c_{FA}} \tag{33}$$

To evaluate correctly the pH of the solution at different temperatures, it is required to know the dependence of both $K_{aFA}$ and $K_w$ on this parameter. $K_w$ is highly temperature dependent, hence increasing with the temperature. Different exhaustive studies have been devoted to this subject [18–20].

The following relation, although has not physical mean, well interpolates the $K_w$ values at different temperatures:

$$K_w = 8.754 \times 10^{-10}\, e^{-\left(\frac{1.01 \times 10^6}{T^2}\right)} \tag{34}$$

Different works have been published on the dissociation constant of HCOOH [21,22] According to Harned and Embree [21] the dissociation constant of HCOOH as a function of temperature can be determined with the following relationship:

$$\lg K_{aFA} = -\left(\frac{173.624}{T}\right) + 17.88348 \lg T - 0.0280397\, T - 39.06123 \tag{35}$$

More recently, Hwa Kim et al. [22] have proposed the following relation:

$$pK_{aFa} = -57.528 + \frac{2773.9}{T} + 9.1232 \lg T \tag{36}$$

that correctly reproduces to a larger extent the dependence of $K_{aFa}$ on the temperature.

In order to evaluate the $c_{H^+}$ concentration for different HCOOH concentrations and different temperatures the following algebraic four equations system, containing both mass and charge balances equations, must be solved:

$$
\begin{aligned}
&(1) \quad \left(c_{H^+} c_{HCOO^-}\right) - K_{aFA}\, c_{FA} = 0 \\
&(2) \quad \left(c_{H^+} c_{OH^-}\right) - K_w = 0 \\
&(3) \quad c_{FA} + c_{HCOO^-} - c_{FA}^0 = 0 \\
&(4) \quad c_{HCOO^-} + c_{OH^-} - c_{H^+} = 0
\end{aligned}
\tag{37}
$$

in which $K_w$ and $K_{aFA}$ can be calculated with the previously reported equations.

## 11. Hydrogen Ion Concentration in the System $H_2SO_4$-Formic Acid

Sulfuric acid is completely dissociated as:

$$H_2SO_4 \rightarrow H^+ HSO_4\text{-} \tag{38}$$

The second dissociation constant (Relation (39)) is equal to $K_{2sulf} = 1.2 \times 10^{-2}$ at 25 °C, that is a dissociation constant greater than the one of formic acid:

$$HSO_4^- \rightleftarrows H^+ + SO_4^{2-} \tag{39}$$

The dependence of this second dissociation constant on the temperature has been studied by three different works respectively published by: (Dickson et al. [23]; Wu and Feng, [24]; Marshall and Jones, [25]). Marshall and Jones determined $K_{2sulf}$ in a very large range of temperature. The data have been interpolated with the relation:

$$\lg K_{2sulf} = 56.889 - 19.8858 \lg T - \frac{2307.9}{T} - 0.006473\, T \tag{40}$$

or alternatively with the more approximated relation:

$$\lg K_{2sulf} = 91.471 - 33.0024 \lg T - \frac{3520.3}{T} \tag{41}$$

Also in this case for evaluating the $H^+$ concentration the following system of algebraic equations, containing both mass and charge balances equations, must be solved:

$$
\begin{aligned}
&(1)\quad (c_{H^+} c_{HSO_4^-}) - K_{1sulf}\, c_{H_2SO_4} = 0\\
&(2)\quad (c_{H^+} c_{HSO_4^-}) - K_{2sulf}\, c_{HSO_4^-} = 0\\
&(3)\quad \left(c_{H^+} c_{OH^-}\right) - K_w = 0\\
&(4)\quad c_{H_2SO_4} + c_{HSO_4^-} + c_{SO_4^=} - c^0_{H_2SO_4} = 0\\
&(5)\quad c_{HSO_4^-} + 2\,c_{SO_4^=} + c_{OH^-} + c_{HCOO^-} - c_{H^+} = 0\\
&(6)\quad \left(c_{H^+} c_{HCOO^-}\right) - K_{aFA}\, c_{FA} = 0\\
&(7)\quad c_{HCOO^-} + c_{FA} - c^0_{FA} = 0
\end{aligned}
\tag{42}
$$

in which $K_w$, $K_{2sulf}$, and $K_{aFA}$ can be calculated with the previously reported equations.

## 12. Hydrogen Ion Concentration in the System H$_3$PO$_4$-Formic Acid

In this case the acidity of the system is the result of the contribution of both $H_3PO_4$ and HCOOH. $H_3PO_4$ gives place to three different dissociation equilibria:

$$H_3PO_4 \rightleftarrows H^+ + H_2PO_4^-\quad K_1 = 7.11 \times 10^{-3} \tag{43}$$

$$H_2PO_{4-} \rightleftarrows H^+ + HPO_4{}^{2-}\quad K_2 = 6.31 \times 10^{-8} \tag{44}$$

$$HPO_{42-} \rightleftarrows H^+ + PO_4{}^{3-}\quad K_3 = 4.8 \times 10^{-13} \tag{45}$$

As it can be seen $K_1$ is comparable with the dissociation constant of formic acid $K_{aFA} = 1.8 \times 10^{-4}$, therefore, the $H^+$ concentration can be calculated rigorously by considering all the equilibrium constants but in particular $K_1$, $K_{aFA}$, and their dependence on the temperature. The dependence of $K_1$ on the temperature was determined by R. Bates [26] for which data were interpolated with the following relationship:

$$-\lg K_1 = \frac{799.31}{T} - 4.5535 + 0.013486\, T \tag{46}$$

C.A. Vega et al. [27] have determined both $K_1$ and $K_2$ as a function of the temperature and data determined have been fitted with the relation:

$$pK_1 = \frac{A}{T} + B + C \lg T \tag{47}$$

The parameters A, B, and C are reported in their work.

However, the equilibrium constants $K_2$ and $K_3$ are too small and their dependence on the temperature can be neglected, so we have considered only the dependence of $K_1$ and $K_{aFA}$ adopting respectively the relations (46) and (35).

To obtain the $c_{H^+}$ concentration in a solution containing $H_3PO_4$ and HCOOH requires to solve the following system of eight algebraic equations containing both mass and charge balances equations:

$$
\begin{aligned}
&(1)\quad (c_{\mathrm{H^+}}\, c_{\mathrm{H_2PO_4^-}}) - K_1 c_{\mathrm{H_3PO_4}} = 0 \\
&(2)\quad (c_{\mathrm{H^+}}\, c_{\mathrm{HPO_4^=}}) - K_2 c_{\mathrm{H_2PO_4^-}} = 0 \\
&(3)\quad (c_{\mathrm{H^+}}\, c_{\mathrm{PO_4^{3-}}}) - K_3 c_{\mathrm{HPO_4^=}} = 0 \\
&(4)\quad (c_{\mathrm{H^+}}\, c_{\mathrm{OH^-}}) - K_{\mathrm{w}} = 0 \\
&(5)\quad c_{\mathrm{H_3PO_4}} + c_{\mathrm{H_2PO_4^-}} + c_{\mathrm{HPO_4^=}} + c_{\mathrm{PO_4^{3-}}} - c^0_{\mathrm{H_3PO_4}} = 0 \\
&(6)\quad c_{\mathrm{H_2PO_4^-}} + 2\,c_{\mathrm{HPO_4^=}} + 3\,c_{\mathrm{PO_4^{3-}}} + c_{\mathrm{OH^-}} + c_{\mathrm{HCOO^-}} - c_{\mathrm{H^+}} = 0 \\
&(7)\quad (c_{\mathrm{H^+}}\, c_{\mathrm{HCOO^-}}) - K_{\mathrm{aFA}}\, c_{\mathrm{FA}} = 0 \\
&(8)\quad c_{\mathrm{HCOO^-}} + c_{\mathrm{FA}} - c^0_{\mathrm{FA}} = 0
\end{aligned}
\tag{48}
$$

## 13. Simulation of the Runs Performed in the Presence of Sulfuric Acid

The runs performed in the presence of sulfuric acid have already been described. The adopted operative conditions of the runs considered are summarized in Table 1. As it can be seen, 12 runs have been performed by changing: the stirring rate, the sulfuric acid concentration, the temperature, the formic acid concentration, and the hydrogen peroxide concentration. In the presence of sulfuric acid as a catalyst, the contribution of formic acid to the protonic concentration is practically null and the pH of the solution changes very little along the time. Moreover, the contribution of $r_{11}$ to the ring opening rate is negligible and we can put $r_{11} \simeq 0$. All the 12 runs of Table 1, made in the presence of sulfuric acid, were simulated with the described model and the obtained agreements are reported in Figures 1–4, while, the kinetic parameters giving the best fitting are reported in Table 4.

**Table 4.** Kinetic parameters and initial protonic concentration used for simulating the ring opening reaction (ROR) runs performed in the presence of sulfuric acid.

| Run | $k_{10}$ ($\times 10^6$) | $\beta$ ($\times 10^3$) | $C^0_{\mathrm{H^+}}$ |
|---|---|---|---|
| 1 | 0.656 ± 0.013 | 6.06 ± 0.12 | 0.1675 |
| 2 | 0.929 ± 0.018 | 15.15 ± 0.30 | 0.1675 |
| 3 | 0.606 ± 0.011 | 0.50 ± 0.02 | 0.1675 |
| 4 | 0.545 ± 0.010 | 0.13 ± 0.01 | 0.1675 |
| 5 | 1.111 ± 0.022 | 5.05 ± 0.11 | 0.1660 |
| 6 | 1.939 ± 0.037 | 5.05 ± 0.11 | 0.1658 |
| 7 | 0.990 ± 0.019 | 3.23 ± 0.06 | 0.3258 |
| 8 | 1.010 ± 0.020 | 2.22 ± 0.05 | 0.0869 |
| 9 | 0.656 ± 0.013 | 2.42 ± 0.05 | 0.1667 |
| 10 | 0.636 ± 0.012 | 3.03 ± 0.02 | 0.1667 |
| 11 | 1.515 ± 0.030 | 2.82 ± 0.05 | 0.1450 |
| 12 | 0.626 ± 0.011 | 3.83 ± 0.04 | 0.1730 |

$k_1$, $k_2$, and $K_{\mathrm{eq}}$ have been imposed on the basis of the relations (25)–(27) and the parameters of Table 2, already reported in our previous work [13]. $k_{10}$ and $\beta$ have been obtained by regression analysis on the experimental data, while, the $\alpha$ values have been roughly estimated from all the runs made resulting approximately: $\alpha_1 = 40$ and $\alpha_2$, $\alpha_3$, and $\alpha_4$ all equal to about 1. These values show the greater nucleophilic power of formic acid with respect to the other molecules ($H_2O$, $H_2O_2$, HCOOOH) and have been estimated considering all the runs made also in the presence of phosphoric acid and HCOOH alone.

Runs 1–4 have been performed all at 70 °C, in the same operative conditions changing only, the stirring rate, 250 rpm for run 4500 rpm for run 31,000 rpm for run 2 and 750 for run 1. $k_{10}$ resulted about the same for runs 1, 3, and 4 with an average value of $6 \times 10^{-7}$, while, run 4 requires a somewhat greater but reasonable parameter of $9.2 \times 10^{-7}$. We can conclude that for these runs the rate changes for the effect of mass transfer, changing the interface area with the stirring speed. As a matter of fact,

the $\beta$ parameter changes from $1.3 \times 10^{-4}$ at 250 rpm to $5 \times 10^{-4}$ at 500 rpm, $6 \times 10^{-3}$ at 750 rpm and $1.5 \times 10^{-2}$ at 1000 rpm. All the other runs have been performed at 750 rpm.

Run 1, 5, and 6 have been made at different temperatures of respectively 70, 80, and 90 °C. Again $k_1$, $k_2$, and $K_{eq}$ have been imposed on the basis of the relations (25)–(27) and parameters of Table 2. On the contrary, $k_{10}$ and $\beta$ have been obtained by regression analysis on the experimental data. $\alpha_1$ has been kept constant equal to 40 and $\alpha_2$, $\alpha_3$, and $\alpha_4$ all equal to 1 giving only to $k_{10}$ the possibility to change with the temperature. Considering the small number of runs available this approximation is reasonable. As it can be seen, $k_{10}$ correctly increases with the temperature, while, the mass transfer parameter remains approximately constant. The dependence of $k_{10}$ can be calculated from the slop of the Arrhenius plot related to the runs 1, 5, and 6 that resulted quite linear. The activation energy resulted $\Delta E_{10}$ = 55,000 ± 2500 Joule/mole.

Runs 7, 8, and 9 have been performed in the same conditions changing only the amount of sulfuric acid added to the aqueous solution that is 1.29 g for run 7, 0.32 g for run 8, and 0.65 for run 9. Moreover, run 9 is identical to run 1. The simulations of these runs show a little change of $k_{10}$ and $\beta$ value and this confirms that H$^+$ concentration strongly affects the ring opening reaction rate and that the adopted model is correct. The small observed differences can be due to the interface area that is the other factor influencing the reaction rate.

Finally, runs 10, 11, and 12 have been made in the same conditions changing only the formic acid concentration. As we have already seen, HCOOH concentration has a strong effect on the ring opening reaction rate. This observation imposes to differentiate its contribution as the most efficient nucleophilic component. We estimated that the nucleophilic power of HCOOH is at least 40 times that of the other components. This difference is satisfactory expressed by the $\alpha_1$ parameter. Run 10 containing 0.65 g of HCOOH (79 wt.%) was performed in the identical conditions as run 1 and 9 and is well simulated with very similar parameters. Run 12 was performed in the presence of 2.69 g of HCOOH and is correctly simulated with parameters similar to the ones of run 10. Run 11, containing an excess of HCOOH corresponding to 10.76 g, gives a $k_{10}$ value that is about double of the previous one. This probably could be due to the fact that we neglected the possibility of the occurrence of the ring opening reaction also inside the oil bulk and hence correlated with the component solubilities although, according to Campanella and Baltanás (Campanella and Baltanás, 2008), the formic acid dissolved in oil is only 3–4% of the total amount. However, by averaging the obtained kinetic parameters in the optimal fitting of all the different performed runs we can write finally:

$$k_{10} = (6.56 \pm 0.27) \times 10^{-7} \exp\left[\left(\frac{-55{,}000 \pm 2500}{8.314}\right)\left(\frac{1}{T} - \frac{1}{343}\right)\right] (\text{L}^2/(\text{Mol}^2 \text{ s})) \qquad (49)$$

## 14. Simulation of the Runs Performed in the Presence of Phosphoric Acid

A set of nine different kinetic runs was made in the presence of phosphoric acid for testing the effect respectively: stirring rate, phosphoric acid concentration, temperature, formic acid concentration, and hydrogen peroxide concentration. All the mentioned runs have been simulated and the parameters giving the best fitting are reported in Table 5.

**Table 5.** Kinetic parameters and initial protonic concentration used for simulating the runs performed in the presence of phosphoric acid.

| Run | $k_{10}$ ($\times 10^7$) | $k_{11}$ ($\times 10^9$) | $\beta$ ($\times 10^4$) | $C_{\text{H}^+}^0$ |
|---|---|---|---|---|
| 13, 14, 15 | 1.62 ± 0.04 | 4.04 ± 0.09 | 1.00 ± 0.02 | 0.0343 |
| 16 | 1.60 ± 0.04 | 4.74 ± 0.09 | 1.01 ± 0.02 | 0.0455 |
| 17 | 1.58 ± 0.04 | 4.04 ± 0.09 | 1.01 ± 0.02 | 0.0370 |
| 18 | 1.64 ± 0.05 | 11.61 ± 0.25 | 1.01 ± 0.02 | 0.0372 |
| 19 | 0.86 ± 0.03 | 2.52 ± 0.06 | 1.01 ± 0.02 | 0.0344 |
| 20 | 5.55 ± 0.95 | 6.16 ± 0.12 | 1.01 ± 0.02 | 0.0343 |
| 21 | 1.62 ± 0.04 | 4.54 ± 0.09 | 40.4 ± 0.9 | 0.0365 |

Again $k_1$, $k_2$, and $K_{eq}$ were imposed as already explained. Runs 13, 14, and 15 were performed in the same conditions with the exclusion of the stirring rate that was changed from 250 to 500 and 750. We have already seen that in the presence of phosphoric acid the stirring rate from 250 to 750 rpm has poor influence on the ESBO concentration profile. This probably means that phosphoric acid strongly interacts with the oxygen of the oxirane ring through the hydrogen bonds so giving place to an adduct with surfactants properties that favor the formation of the oil–water emulsion with a high interface area also at low stirring rate. The parameters are, therefore, almost the same for all these three runs. Run 14, 16, and 17 have been made in the same conditions, changing only the $H_3PO_4$ amounts that were respectively 0.65 g, 1.3 g, and 0.79 g (w = 80%). Surprisingly, the effect of phosphoric acid concentration on the ring opening reaction rate is poor and the parameters for the best fitting are practically the same. This behavior is probably due to the small difference in $H^+$ concentration in the mentioned runs changing from 0.0343 to 0.0455 mol/L and to the not negligible intervention of the mechanism of ring opening independent of the pH based on the nucleophilic attack instead of the protonic one.

Run 18 must be compared with run 14, because, it has been made in the same conditions with the exclusion of the formic acid amount that was 5.4 g (w = 79%) for the run 14 and 10.76 g for the run 18. $k_{10}$ would be the same in both cases, while, $k_{11}$ resulted somewhat different, that is, for obtaining a satisfactory simulation of the run 18 $k_{11}$ must be taken 2.875 greater than run 14. A possible explanation could be that at high HCOOH concentration the ring opening reaction consistently occurs also in the oil phase, but to demonstrate this assumption many other experiments are required. Runs 14, 19, and 20 have been made in the same conditions but changing the temperature respectively at 70, 60, and 80 °C. Both the parameters $k_{10}$ and $k_{11}$ increase with the temperature and the activation energies have been determined from the slopes of the Arrhenius plots related to $k_{10}$ and $k_{11}$ that both have given a good linear trend. Run 21 occurs in a full chemical regime, because, it requires a very high value of $\beta$ to be correctly simulated.

At last by averaging the kinetic parameters found for all the performed runs we can write:

$$k_{10} = (1.62 \pm 0.11) \times 10^{-7} \exp\left[\left(\frac{-88,000 \pm 3930}{8.314}\right)\left(\frac{1}{T} - \frac{1}{343}\right)\right] (L^2/(Mol^2 \, s)) \tag{50}$$

$$k_{11} = (4.24 \pm 0.35) \times 10^{-9} \exp\left[\left(\frac{-45,000 \pm 1980}{8.314}\right)\left(\frac{1}{T} - \frac{1}{333}\right)\right] (L^2/(Mol^2 \, s)) \tag{51}$$

*Simulation of the Runs Performed in the Absence of Mineral Acids*

As it has been previously seen (Table 1 and Figure 11) three kinetic runs have been performed in the absence of mineral acid. In one run, run 23, all the usual reaction components (HCOOH, $H_2O$, $H_2O_2$), were added to the epoxidized soybean oil, while, in other two runs only HCOOH and $H_2O$ were added. Obviously, in the two last cases the formation and decomposition of HCOOOH do not occur, the pH remains constant, and only the ring opening reaction occurs. These runs were also simulated and the parameters giving the best fitting are reported on Table 6.

**Table 6.** Kinetic parameters used for simulating the runs performed in the absence of mineral acids.

| Run | $k_{10}$ ($\times 10^7$) | $k_{11}$ ($\times 10^9$) | $\beta$ ($\times 10^4$) | $C_{H^+}^0$ |
|---|---|---|---|---|
| 22 | 1.61 ± 0.03 | 2.02 ± 0.04 | 1.01 ± 0.02 | 0.0558 |
| 23 | 1.81 ± 0.04 | 2.52 ± 0.04 | 2.02 ± 0.04 | 0.0188 |
| 24 | 1.51 ± 0.03 | 2.12 ± 0.04 | 1.72 ± 0.03 | 0.0180 |

In run 23, $k_1$ and $k_2$ and $K_{eq}$ were imposed as in the previous cases. $\alpha_1$ and $\gamma_1$ were imposed, too, while, $k_{10}$ and $k_{11}$ were determined by regression analysis on the experimental data taking into account, when necessary, also of the change of $H^+$ concentration occurring as a consequence of the HCOOOH decomposition. The interpretation of run 22 and 24 required to change somewhat the program code,

because, in the absence of $H_2O_2$ the reactions of HCOOOH, formation and decomposition do not occur. These runs are important because they show the influence of HCOOH in the ring opening reaction without the interference of other occurring reactions. As it can be seen, the obtained parameters are in satisfactory agreement with both run 23 and the runs made in the presence of phosphoric acid. Very important is the result of run 22 that demonstrate the greatest nucleophilic power of HCOOH.

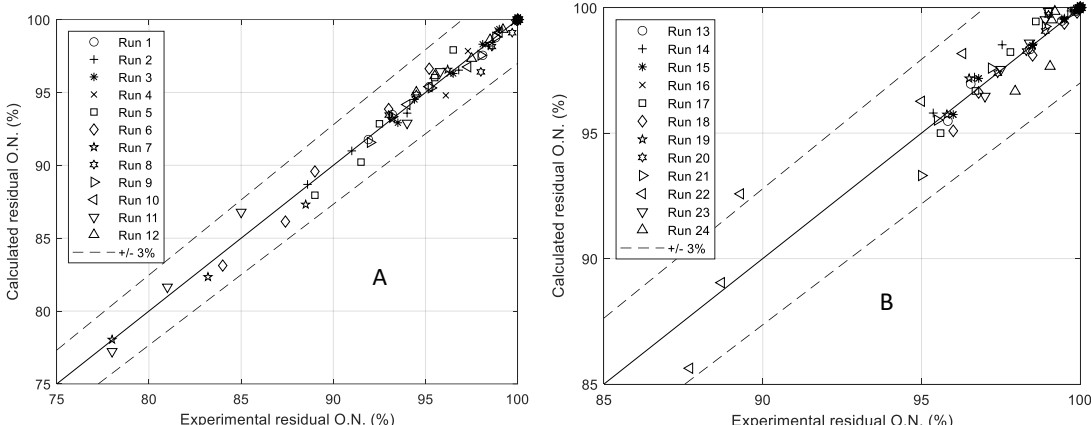

**Figure 11.** (**A**,**B**) A is the parity plot related to all the runs performed in the presence of $H_2SO_4$ (runs 1–12). B is the parity plot related to all the other performed runs (runs 13–24 of Table 1).

In conclusion, the developed model for interpreting the ring opening reaction rate, in different operative conditions, despite the difficulty of determining with precision all the parameters of the model, has given a satisfactory performance interpreting almost all the runs made. The obtained agreements in the simulation of the runs, characterized by the presence of a high HCOOH concentration, is poor requiring more experimental runs for a more correct interpretation.

Finally, in Figure 11A,B two parity plots compare all the experimental data with the calculated ones. As it can be seen the performance of the model is quite satisfactory giving place to an average error of less than 3%.

Finally, in Table 7, some statistical parameters are reported to better illustrate the quality of the overall fit for, respectively, the runs with sulfuric and phosphoric acid.

**Table 7.** Statistical parameters for the overall fit.

| Parameter | Runs with Sulfuric Acid | Runs with Phosphoric Acid |
|:---:|:---:|:---:|
| $R^2$ | 0.9863 | 0.9203 |
| $R^2_{adj}$ | 0.9861 | 0.9192 |
| Mean error (%) | 0.45 | 0.42 |

### 15. Conclusions

Many different kinetic runs have been made for studying the kinetics of the oxirane ring opening reaction in the presence of respectively sulfuric acid, phosphoric acid and in the absence of mineral acid starting, in all cases, from fully epoxidized soybean oil. The scope was to investigate about the role of acidity on the oxirane ring opening reaction rate or the eventual effect on the rate of the nucleophilic character of both reactants and products. It has been shown that the oxirane ring opening reaction, according to the operative conditions, can occur with two different reaction mechanism one characterized by the protonic attack to the epoxide oxygen, occurring at the liquid–liquid interface, and another one by an attack of a nucleophile to one of the two carbon atoms of the epoxide ring. Two different kinetic laws can be derived from the mentioned reaction mechanisms.

A biphasic kinetic model was developed for interpreting all the performed kinetic runs determining the related parameters by regression analysis on all the experimental data. In any case, also the contribution of the mass transfer was considered. This contribution is due to the migration of the bulky epoxide molecules from the oil bulk to the water–oil interface and is characterized by the mass transfer coefficient β and the epoxide gradients calculated for any single kinetic run. Despite the large number of imposed parameters recovered by previously published works the obtained agreements in simulating all the runs are satisfactory.

**Author Contributions:** E.S. coordinated all the aspect of the work, R.T. (Rosa Turco) has made the experimental runs, V.R. and R.T. (Riccardo Tesser) contributed in the elaboration of the kinetic model and calculations, M.D.S. helped in the coordination of all the aspects of the work. All authors have read and agreed to the published version of the manuscript.

**Funding:** Thanks are due to Desmet Ballestra Co. and Eurochem Engineering Ltd. for funding the research.

**Conflicts of Interest:** The authors declare no conflict of interest.

## Glossary

**List of symbols**

| | |
|---|---|
| $c_i$ | Concentration identified by the subscript i (mol/l) in aqueous phase, (mol/cm$^3$ of oil) in oil phase |
| $c_i^0$ | Initial concentration of i |
| $c_e$ | Concentration of EPOX in the oil liquid bulk (mol/cm$^3$ of oil) |
| $c_f$ | Concentration of EPOX at the liquid-liquid interface (mol/cm$^3$ of oil) |
| $\Delta E_1$ | Activation energy of performic acid formation (Joule/mol) |
| $\Delta E_2$ | Activation energy of performic acid decomposition (Joule/mol) |
| $\Delta H$ | Enthalpy change of the performic acid formation (Joule/mol) |
| $k$ | Generic kinetic constant |
| $k_{1\text{-}323}$ | Kinetic constant at 323 K of performic acid formation (l$^2$/mol$^2$ s) |
| $k_{2\text{-}323}$ | Kinetic constant at 323 K of performic acid decomposition (1/s) |
| $K_{eq}$ | Equilibrium constant of performic formation |
| $K_{eq,ref}$ | Equilibrium constant of performic formation at a reference temperature |
| $K_1, K_2, K_3$ | Equilibrium constants of the three ionic dissociation of $H_3PO_4$ |
| $K_{aFA}$ | Equilibrium constant of the ionic dissociation of formic acid |
| $K_{1sulf}, K_{2sulf}$ | First and second ionic dissociation equilibrium constants of sulfuric acid |
| $K_w$ | Ionic product of water |
| $r$ | Generic reaction rate |
| $r_1$ | Rate of performic acid formation (mol/l s). Volume referred to the aqueous solution |
| $r_2$ | Rate of performic acid decomposition (mol/l s). Volume referred to the aqueous solution |
| $r_{10}$ | Rate of epoxidation reaction promoted by acids (mol/(cm$^3$ s)). Volume referred to the oil solution |
| $r_{11}$ | Rate of epoxidation reaction promoted by a nucleophile (mol/(cm$^3$ s)). Volume referred to the oil solution |
| $r_g$ | Overall epoxidation rate (mol/(cm$^3$ s)). Volume referred to the oil solution |
| rpm | Stirring rate, Number of revolutions of the stirrer per minute |
| $r_t$ | ESBO mass transfer rate (mol/(cm$^3$ s)). Volume referred to the oil solution |
| $t$ | Time (s) |
| $T$ | Temperature (K) |

**Greek Letters**

| | |
|---|---|
| $\alpha_N$ | Ring-opening parameters of nucleophilic power, $N = 1$ (FA), $N = 2$ ($H_2O_2$), $N = 3$ (PFA), $N = 4$ ($H_2O$) |
| $\gamma_N$ | Ring-opening parameters of nucleophilic power, $N = 1$ (FA), $N = 2$ ($H_2O_2$), $N = 3$ (PFA), $N = 4$ ($H_2O$) |
| $\beta$ | Liquid-liquid mass transfer coefficient, (1/s)) |

**Subscripts defining concentrations**

| | |
|---|---|
| EPOX | Epoxide |
| $EPOX^+$ | Protonated epoxide |
| Nu | Generic nucleophilic component |
| FA | Formic acid |
| PFA | Performic acid |
| $H_2O$ | Water |
| $H_2O_2$ | Hydrogen peroxide |
| $H_2SO_4$ | Sulfuric acid |
| $HSO_4^-$ | First dissociation sulfuric anion |
| $SO_4^=$ | Second dissociation sulfuric anion |
| $H_3PO_4$ | Phosphoric acid |
| $H_2PO_4^-$ | First dissociation phosphoric acid anion |
| $HPO4^=$ | Second dissociation phosphoric acid anion |
| $PO_4^{3-}$ | Third dissociation phosphoric acid anion |
| $HCOO^-$ | Dissociation of formic acid anion |
| $OH^-$ | Hydroxyl |
| $H^+$ | Proton |

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
