# Peer review of "Soybean Oil Epoxidation: Kinetics of the Epoxide Ring Opening Reactions"

_processes, doi:10.3390/pr8091134_

Round 1

Reviewer 1 Report

This article deals with the study of the epoxidation of soybean oil. The authors present an exhaustive study on the process in multiple operating conditions, shedding new light on the mechanism of the process. In my opinion, the authors provide experimental results that satisfactorily support their conclusions. For all these reasons, I see no reason not to recommend the publication of the article in its present form.

Author Response

We thank very much Reviewer 1 for the positive evaluation of our work

Reviewer 2 Report

See attached PDF-file

Reviewer 3 Report

See an attachment
